# Evaluation of the association between postural control and sagittal curvature of the spine

**Arkadiusz Łukaz Żurawski**[1,2]*, **Wojciech Piotr Kiebzak**[1,2], **Ireneusz M. Kowalski**[3], **Grzegorz Śliwiński**[1,4], **Zbigniew Śliwiński**[1]

**1** Institute of Health Sciences, Collegium Medicum, The Jan Kochanowski University, Kielce, Poland, **2** Świętokrzyskie Centre for Paediatrics, Provincial Integrated Hospital, Kielce, Poland, **3** University of Warmia and Mazury in Olsztyn, Olsztyn, Poland, **4** Institute of Biomedical Engineering, TU Dresden, Dresden, Germany

* azurawski@onet.eu

**Citation:** Żurawski AŁ, Kiebzak WP, Kowalski IM, Śliwiński G, Śliwiński Z (2020) Evaluation of the association between postural control and sagittal curvature of the spine. PLoS ONE 15(10): e0241228. https://doi.org/10.1371/journal.pone.0241228

**Data Availability Statement:** All relevant data are within the paper and its Supporting Information files.

## Abstract

### Introduction

Balance is key to controlling body posture. Balance is typically assessed by measures of the body's vertical orientation, obtained by balancing out the forces acting on different body segments. The ability to maintain balance is assessed by evaluating centre of pressure (CoP) displacement; such assessments are typically used to evaluate responses to a treatment process.

### Purpose of study

This study evaluated the efficiency of compensatory reactions in children according to the extent of thoracic kyphosis and lumbar lordosis.

### Materials and method

The study enrolled 312 children aged 8–12 years, including 211 patients with postural disorders: thoracic kyphosis outside the 47–50-degree range and lordosis outside the 38–42-degree range (study group). A control group was also recruited and comprised 101 children without postural disorders. The DIERS formetric 4D system was used to assess posture and CoP displacement.

### Results

Children in the study group showed a significantly greater range of CoP displacement than children in the control group. The kyphosis angle correlated with the maximum CoP displacement in the coronal plane and the maximum CoP displacement in the sagittal plane during gait. The kyphosis angle also correlated with the maximum CoP displacement back in the static test. The size of the lordosis angle correlated with the maximum displacement of CoP in the coronary plane during gait, and with the maximum displacement of CoP toward the left, forward, and backward in the static test. The correlation coefficient of the lordosis angle with displacement of the CoP in the sagittal plane was 0.999.

**Funding:** Project financed under the program of the Minister of Science and Higher Education called "Regional Initiative of Excellence" in the years 2019-2022, project no. 024/RID/2018/19, amount of financing 11 999 000,00 zł. The funders had no role in study design, data collection and analysis, decision to publish, or preparation of the manuscript.

**Competing interests:** The authors have declared that no competing interests exist.

## Conclusions

1. We found an association between kyphosis and lordosis and the amplitude of CoP displacement, which may reflect the postural control system's response to biomechanical destabilisation caused by changes in kyphosis and lordosis.

2. The lordosis angle correlation strength for displacement of CoP in sagittal plane is 0.999 and adopts a linear value.

## Introduction

Posture defines the individual placement of the body's limbs in relation to one another under dynamic and static conditions while in an upright position. It corresponds to the typical biomechanical conditions for the functioning of the human body [1]. Correct body posture allows one to retain a vertical position and limits movement of the centre of mass in relation to the support plane [2, 3], ensuring maximum stability with the minimum muscle involvement [4]. The amplitude of thoracic kyphosis and lumbar lordosis, which stabilize the spine, preventing curvature in other planes, are important for maintaining a correct posture [5].

A faulty posture comprises a set of irregularities, defined as small, individual deviations from the correct posture, which can be corrected using appropriate therapy [6, 7]. There are many causes for posture disorders, including limb-length disturbances, which change the position of the pelvis and thus affects the shape of the spine [8–11]. In such cases, limb length equalization is sufficient intervention [8, 12]. Asymmetrical shoulder loading also affects the shape of the spine [13]. Posterior and pelvic muscle insufficiency is another source of posture disorders, and in these cases therapy improving the strength of these muscles is effective for correcting the disorders [14, 15]. Many studies have shown that spinopelvic alignment and sagittal balance are major contributors to an energy-efficient posture of the individual in the healthy and diseased states [16–18].

Body posture disorders are common among the paediatric population and are increasingly becoming a serious epidemiological concern in this age group [6, 7, 19, 20]. The literature indicates that determining the angle of healthy curvature of the spine in the sagittal plane is challenging, with discrepant findings across studies. In radiological studies, in a population of individuals aged 20–63 years, the mean angle of thoracic kyphosis was 28.9˚ ± 12.1˚ in women and 31.2˚ ± 7.9˚ in men, while the average angle of lumbar lordosis was 45.7˚ ± 12.9˚ in woman and 43.9˚ ± 10.8˚ in men [21]. The range of a healthy thoracic kyphosis angle should be 20–40˚ in growing teenagers, based on data from 1960 provided by the Scoliosis Research Society [22]. Using radiological examination, Lin et al. determined the angle of lumbar lordosis at an average age of 50 years to be 33.2˚ ± 12.1˚; they did not observe significant differences between women and men [23]. Radiological examinations of 350 healthy people aged 18–50 years showed average values of 45˚ for lumbar lordosis [24]. In radiological studies of children aged 3–10 years, the average thoracic kyphosis angle was 42.0˚ ± 10.6˚ and the average lumbar lordosis angle was 53.8˚ ± 12.0˚ [25]. In the same study, the mean thoracic kyphosis angle was 45.8˚ ± 10.4˚ and lumbar lordosis angle was 57.7˚ ± 11.1˚ for the age range 10–18 years [25]. Further radiological examinations in children aged 8–19 years reported an average thoracic kyphosis angle of 47.47˚ ± 12.7˚ and average lumbar lordosis angle of 39.6˚ ± 12.4˚ [26]. A study of people aged between 2 and 27 years, conducted by Wenger and Frick, reported that the average angle of thoracic kyphosis increased with age, from 20˚ in childhood, to 25˚ in

adolescence, to 40˚ in adulthood, which makes it challenging to establish uniform norms concerning changes occurring during puberty [22]. The DIERS system provides norms for certain parameters that are considered physiological: a kyphotic angle of 48˚ ± 9˚ and a lordosis angle of 43˚ ± 8˚ for women and 36˚ ± 7˚ for men [27]. Thus, the lack of consistent values in the literature that define a healthy spinal curve in the sagittal plane impedes the comparison of studies, and may be ascribed to varying inclusion criteria, which makes the distinction between the upper boundary of standard thoracic kyphosis and severe juvenile deformation almost impossible [5].

Kyphosis and lordosis influence balance in the upright individual. Balance allows one to regain a previous state of equilibrium in the body while performing or after the completion of motor tasks [28]. The determinant used to define the ability to maintain balance is the centre of pressure (CoP) displacement; evaluating CoP displacement is used to reflect responses to treatment. An accurate definition of the parameters used to describe balance allows one to choose appropriate actions to improve balance via therapy, and knowledge of changes occurring within the parameters describing balance dysfunction allows appropriate selection of treatment methods to optimize the therapeutic process. Spontaneous CoP displacement consist of two-dimensional information (observation of individual elements of the route on a statokinesiogram), i.e., analysing postural shifts in the sagittal and coronal plane separately allows one to determine postural stability [28–30]. Additionally, gait analysis in relation to body posture also yields useful information. Changes in the body shape cause an asymmetrical load on the legs, which generates a postural muscle imbalance as well as a displacement in the CoP [31]. CoP displacement during quiet standing reflects the net contribution of the central nervous system (CNS) to control the movement of the Centre of Mass (CoM). Where the horizontal acceleration of the CoM is proportional to the difference between the CoP-CoM [32]. During quiet stance it is the hip abductors/adductors that generate torque to control ML movement, and it is the ankle plantar flexors/dorsiflexors that control A/P movement. During walking, the movement of the CoP moves from posterior to anterior under the foot, and M/L. Although the above concept holds true (ie., CoP reflects net output of CNS control), how it does this is more complex given the multiple degrees of freedom, and joint motions.

With the increase in postural dysfunction, the path covered by the CoP in a given unit of time is extended, indicating that compensatory reactions are weakened [32]. When the CNS senses that this shift in the centre of gravity (CoG) requires correction, the CoP changes until it lies posterior to the CoG, allowing the body to return to it's original condition. Assessment of the changes in the CoG and CoP conditions has demonstrated that the plantar flexors-dorsiflexors that control the net ankle moment can regulate the body's CoG [32]. The CNS attempts to counteract gravitational forces on the body by adjusting the alignment of body segments, so that any disorder in this biomechanical system will reduce the efficiency of equivalent reactions. The net muscle moment is the sum of the balance and postural components [32]. There is a significant relationship between body posture, the efficiency of compensatory reactions, and the gait quality [33]. In cases of minor deformations this may mainly apply to the sagittal plane, but with pronounced abnormal curvatures of the spine, dysfunctions may also involve movement of the CoP in the coronal plane [34].

## Purpose of study

This study investigated the influence of thoracic kyphosis and lumbar lordosis on compensatory reactions in children without posture disorders and in those with pathological thoracic kyphosis and lumbar lordosis angles.

## Materials and method

### Study participants

The study involved 312 children aged 8–12 years. Study participants were recruited consecutively from among patients who came to our outpatient clinic and included children who met the inclusion criteria.

The study group consisted of 211 patients with postural disorders, in terms of the degree of thoracic kyphosis and lumbar lordosis: kyphosis outside the 35–59˚ range and lordosis outside the 27–51˚ range [26]. This group comprised 112 girls (53.1%) and 99 boys (46.9%). The control group consisted of 101 children in whom no postural disorders were found in a clinical trial. Children in this group were recruited consecutively from among those who came for a preventive body posture examination. The children included in the control group did not show any signs of body posture disorder in physiotherapeutic assessment. The group comprised 51 girls (50.5%) and 50 boys (49.5%). The morphological parameters of children from both groups are presented in Table 1.

The research was conducted from 2016 to 2018. The study was approved by the Bioethics Committee of the Faculty of Medicine and Health Sciences, Jan Kochanowski University, Kielce (consent no. 1/2016 issued on 15 January 2016). Informed consent for participation in the study was obtained from each child and their parents/guardians. Participation in the research was voluntary, combined with ensuring anonymity regarding the Personal Data Protection Act of 29.08.1997 (Journal of Laws No. 133, item 883. Republic of Poland).

### Inclusion criteria

The inclusion criteria were as follows: age 8–12 years; good overall condition (Eastern Cooperative Oncology Group performance status scale $\leq 2$); ability to perform at least personal activities; consent from the legal guardian/parent to participate in the research. Additionally, the study group participants were included if they had a body posture defect: on the DIERS system, a posture defect was defined as a pelvic inclination $> 5$ mm, lateral deviation $< 5$ mm, and surface rotation $< 5$ degrees. If the lateral deviation was $> 5$ mm and the surface rotation was $> 5$ degrees, with a pelvic inclination of $< 5$ mm, scoliosis was diagnosed [35]. Furthermore, control group participants were included if there was no postural defect.

### Exclusion criteria

Children were excluded from participation if there were any co-morbidities that may affect the body axis, e.g., Sheuerman disease, genetic diseases, such as Beckwith–Wiedemann syndrome, or metabolic diseases, or if they had a body mass index below 16.99 or above 29.99.

### Research methodology

Body posture was assessed under static conditions in a habitual standing position, with the back to the camera and facing straight ahead. Body posture assessment was accompanied by

**Table 1. Morphological parameters of children from the study group and the control group.**

| Parameter | Study group | | Control group | |
|---|---|---|---|---|
| | Average | SD | Average | SD |
| Age | 10.7 | 1.3 | 10.7 | 1.4 |
| Body height | 1.4 | 0.2 | 1.4 | 0.1 |
| Body mass | 34.7 | 11.9 | 38.7 | 8.0 |
| BMI | 20.2 | 2.4 | 20.0 | 2.5 |

simultaneous measurement of ground reaction force to the feet, and the CoP movement in a static position.

The DIERS formetric 4D system (DIERS International Gmbh, Schlangenbad, Germany) was used to assess posture [36–37]. Using rasterstereography, the DIERS formetric 4D device allows for photogrammetric video recording of the subject's back surface. By analysing the surface of the back, the data collected allow analysis of the spinal axis [38]. The digital recording lasted about 3 seconds and comprised 12 photos; data analysis was performed using the built-in computer software.

Measurements of ground reaction forces on the feet and the CoP movement in a static position were conducted using the DIERS pedoscan device in combination with the DIERS formetric system [39]. The subject stood in the centre of a measuring platform of $80 \times 100$ cm. The feet were positioned straight ahead in their natural, relaxed position. The maximum peak in four directions (left, right, forward, and backward) was evaluated. The measurement was carried out for 3 seconds at the same time as the shape of spine was assessed with the DIERS formetric system.

The analyses of ground reaction forces and equivalent reactions under dynamic conditions were conducted using a DIERS pedogait device consisting of a treadmill and a built-in pedobarographic platform, measuring $80 \times 100$ cm, with 5376 built-in sensors. The measurement was made during a calm walk at a speed of 2 km/h, over a distance of 16 m. The maximum peak in the sagittal and frontal planes was evaluated.

The following parameters were assessed:

**Kyphosis angle.** This is the kyphotic angle measured between Vertebra Prominens and the estimated position of T12.

**Lordosis angle.** This is the lordotic angle, measured between the estimated position of T12 –DM (Centre between Dimple L and Dimple R

**Maximum displacement of centre of pressure in coronal plane during gait.** The parameter is calculated in millimetres [mm], which is the greatest displacement (maximum peak) of the projection of the center of pressure in the coronal plane during gait at a distance of 16 m.

**Maximum displacement of centre of pressure in sagittal plane during gait.** The parameter is calculated in millimetres [mm], which is the greatest displacement (maximum peak) of the projection of the center of pressure in the sagittal plane during gait at a distance of 16 m.

**Maximum displacement of the CoP in the coronal plane under static conditions.** This parameter is calculated in millimetres [mm] and shows the distance (maximum peak) covered by the projection of the center of pressure on to the ground in 3 seconds in the coronal plane (body to the left and right).

**Maximum displacement of the CoP in the sagittal plane under static conditions.** This parameter is calculated in millimetres [mm] and shows the distance (maximum peak) covered by the projection of the center of pressure on to the ground in 3 seconds in the sagittal plane (forward and backward).

## Statistical analysis

Statistical analysis was conducted using IBM SPSS Statistics v23 software (IMB SPSS Inc, Chicago, IL, USA). Basic descriptive statistics analysis was performed. To assess the normality of the distribution of variables, we used the Kolmogorov–Smirnov test [40]. To establish the relationship between the studied variables, we used the Spearman rank correlation test [41]. The level of significance was set at $p < 0.05$.

## Results

Basic descriptive statistics of the examined quantitative variables were calculated. Analyses were performed separately for the study group and the control group.

Table 2. Comparison of values of examined variables in the study and control groups.

| | study group (n = 211) | | control group (n = 101) | | | | | |
|---|---|---|---|---|---|---|---|---|
| | M | SD | M | SD | U | Z | p | r |
| Angle of thoracic kyphosis VP-ITL [°] | 41,37 | 9,60 | 39,45 | 8,28 | 9695,0 | -1,288 | 0,198 | 0,07 |
| Angle of lumbar lordosis ITL-DM [°] | 37,05 | 9,55 | 38,33 | 8,59 | 9593,0 | -1,425 | 0,154 | 0,08 |
| Maximum displacement of centre of pressure in coronal plane during gait [mm] | 104,63 | 25,08 | 98,73 | 19,22 | 9126,5 | -2,051 | **0,040** | 0,12 |
| Maximum displacement of centre of pressure in sagittal plane during in gait [mm] | 202,07 | 58,32 | 160,39 | 32,37 | 5542,5 | -6,858 | **<0,001** | 0,39 |
| Maximum displacement of the CoP in the coronal plane under static conditions [mm] (to the left) | 1,03 | 1,55 | 0,56 | 0,33 | 6846,5 | -5,109 | **<0,001** | 0,29 |
| Maximum displacement of the CoP in the coronal plane under static conditions [mm] (to the right) | 0,80 | 0,64 | 0,55 | 0,35 | 7445,0 | -4,307 | **<0,001** | 0,24 |
| Maximum displacement of the CoP in the sagittal plane under static conditions [mm] (forward) | 1,18 | 1,01 | 0,83 | 0,41 | 7108,5 | -4,758 | **<0,001** | 0,27 |
| Maximum displacement of the CoP in the sagittal plane under static conditions [mm] (backward) | 0,69 | 0,97 | 0,43 | 0,28 | 8145,5 | -3,367 | **0,001** | 0,19 |

*M*–average; *SD*–standard deviation; *U*—Mann-Whitney U test result; *Z*—standardized value; *p*–relevance; r—effect strength

Most of the examined variables had a non-normal distribution. Hence, we conducted statistical analyses using nonparametric tests. As our inclusion criteria did not allow inclusion of individuals with kyphosis and lordosis below and above the reference values, there was no significant difference between the groups. All parameters describing CoP movement were significantly different between the study and control groups. Details are presented in Table 2.

We then investigated the relationship between body posture and CoP movements under static and dynamic conditions. A series of Spearman's ρ rank correlation analyses were performed separately for the study group and the control group. As the angle of kyphosis increased, the maximum displacement of the CoP in the coronal as well as in the sagittal plane increased (Table 3). The increase in the maximum CoP displacement in the sagittal plane relative to the angle of kyphosis was markedly faster than in the frontal plane. As the angle of kyphosis increased, the maximum backward displacement of the CoP also increased (Table 3).

Along with the increase in the angle of lordosis, an increase in CoP displacement were observed. There was an increase in the maximum displacement of the CoP in the coronal plane, maximum displacement of the CoP in the coronal plane (static conditions, body to the left). Moreover, there was a reduction in the forward as well as in the backward displacement of the maximum CoP in the static test with an increased lordosis angle (Table 4).

The displacement of the CoP in the sagittal plane increased with an increment in the lordosis angle. Interestingly, the results of the tests form a straight trend line (Fig 1).

## Discussion

This study investigated whether the magnitude of thoracic kyphosis and lumbar lordosis has an impact on compensatory reactions in children. We found that the kyphosis angle correlated with the maximum CoP displacement during gait (Table 3), and the lordosis angle correlated with the maximum CoP displacement during gait and with the maximum C0P displacement under static conditions (Table 4). Children with impaired body posture were characterized by significantly higher maximum CoP displacement than children with a physiologically normal spine shape (Table 2), which indicated a reduced ability to control vertical posture.

Our results indicated a relationship between the shape of the spine and the maximum displacement of CoP under static conditions. The kyphosis angle correlated weakly with the maximum backward displacement of the CoP (r = 0.177 p = 0.01). The lordosis angle correlated with the maximum leftward, forward, and backward COP displacement; these correlations were also weak (r = 0.154, -0.17, and 0.381, respectively), although statistically significant (p < 0.05).

**Table 3. Significant relationships between the angle of thoracic kyphosis and other variables tested in the study group and in the control group.**

| | | Angle of thoracic kyphosis VP-ITL [°] | |
|---|---|---|---|
| | | study group | control group |
| Maximum displacement of centre of pressure in coronal plane during gait [mm] | Spearman ρ | **0.182** | -0.043 |
| | relevance | **0.008** | 0.672 |
| Maximum displacement of centre of pressure in sagittal plane during gait [mm] | Spearman ρ | **0.438** | 0.155 |
| | relevance | **<0.001** | 0.121 |
| Maximum displacement of the CoP in the coronal plane under static conditions [mm] (to the left) | Spearman ρ | 0.126 | -0.029 |
| | relevance | 0.068 | 0.770 |
| Maximum displacement of the CoP in the coronal plane under static conditions [mm] (to the right) | Spearman ρ | 0.118 | -0.140 |
| | relevance | 0.089 | 0.164 |
| Maximum displacement of the CoP in the sagittal plane under static conditions [mm] (forward) | Spearman ρ | -0,066 | -0,062 |
| | relevance | 0,341 | 0,541 |
| Maximum displacement of the CoP in the sagittal plane under static conditions [mm] (backward) | Spearman ρ | **0.177** | 0.040 |
| | relevance | **0.010** | 0.688 |

Nalut et al. [42] made similar observations, although their research showed a clearly stronger effect on the path-length in the coronal plane (increased by 50%) than in the sagittal plane (increased by 25%). Wilczyński et al. [43] showed both an increase in the CoG path-length in

**Table 4. Significant relationships between the angle of lumbar lordosis and other variables tested in the study group and in the control group.**

| | | Angle of lumbar lordosis ITL-DM [°] | |
|---|---|---|---|
| | | study group | control group |
| Maximum displacement of centre of pressure in coronal plane during gait [mm] | Spearman ρ | **0.265** | -0.031 |
| | relevance | **<0.001** | 0.762 |
| Maximum displacement of centre of pressure in sagittal plane during gait [mm] | Spearman ρ | **0.999** | 0.129 |
| | relevance | **<0.001** | 0.199 |
| Maximum displacement of the CoP in the coronal plane under static conditions [mm] (to the left) | Spearman ρ | **0.154** | -0.021 |
| | relevance | **0.025** | 0.836 |
| Maximum displacement of the CoP in the coronal plane under static conditions [mm] (to the right) | Spearman ρ | 0.133 | -0.032 |
| | relevance | 0.054 | 0.748 |
| Maximum displacement of the CoP in the sagittal plane under static conditions [mm] (forward) | Spearman ρ | **-0.172** | -0.005 |
| | relevance | **0.012** | 0.964 |
| Maximum displacement of the CoP in the sagittal plane under static conditions [mm] (backward) | Spearman ρ | **0.381** | 0.037 |
| | relevance | **<0.001** | 0.715 |

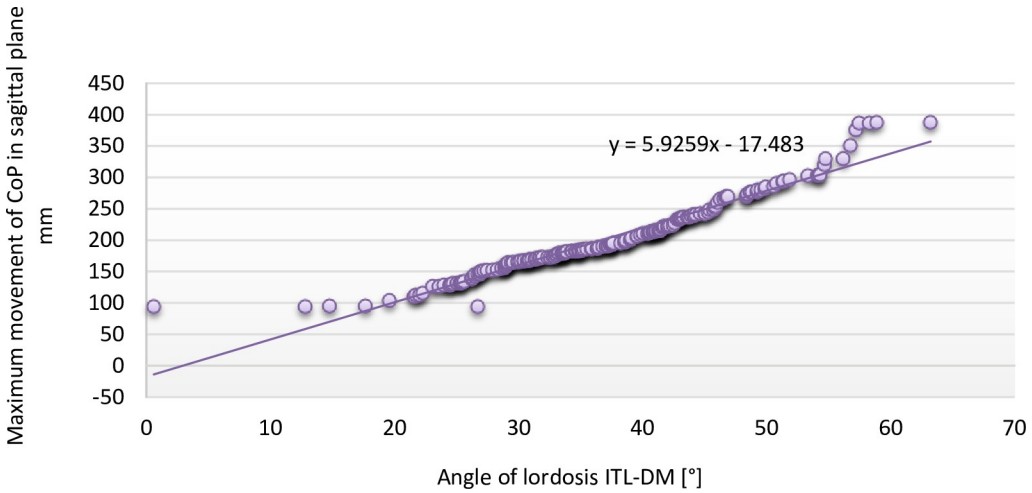

**Fig 1. Relationship of angle of lordosis to maximum displacement of centre of pressure in sagittal plane during gait.**

both planes, as well as an increase in the maximum displacement in all four directions, with the increase in thoracic kyphosis; these findings agreed with the results obtained in the present study. Subsequently, they emphasized that the impact of thoracic kyphosis disorders on the quality of equivalent reactions is independent of the co-occurrence of spinal axis disorders in the coronal plane [44]; this allows us to distinguish this element as independent of postural dysfunction as a whole.

Changing the angle of kyphosis affects the dorsal and calf muscle tension, contributing to the weakening of equivalent reactions [31] Therefore, a change in the angle of kyphosis can start a chain of biokinematic changes. However, these relationships should be verified in future studies involving the simultaneous measurement of kyphosis, lordosis, postural muscle tone, and CoG displacement.

The curvature of the spine had a markeldy larger impact on the maximum displacement of the CoP during gait. The angular kyphosis angle correlated with the maximum displacement of the CoP in the coronal plane during gait with an effect force of 0.182 ($p = 0.008$), and in the sagital plane with an effect force of 0.438 ($p < 0.001$). The lumbar lordosis angle correlated with the maximum displacement of the CoP during gait in the coronal plane with an effect force of 0.265 ($p < 0.001$), and in the sagittal plane with an effect force of 0.999 ($p < 0.001$). Therefore, these correlations were of moderate strength, except for the relationship between the lumbar lordosis angle and the maximum displacement of the CoP in the sagittal plane, which was very strong. This may also be due to an increase in the rear-foot load with the increase in the lumbar lordosis angle, as reported by Suoza et al. [45] who observed this phenomenon in patients with temporomandibular joint dysfunction.

Comparison of the magnitude of the maximum CoP displacement in a group of children with a physiologically normal spine and a group with impaired posture revealed statistically significant differences between the groups in all directions of CoP displacement, both in standing position and while walking, with smaller shifts in children with a physiologically normal spine (Table 2). Particularly interesting observations relate to the strength of the effect of these differences, which is definitely greater than the correlation of individual shifts with the size of thoracic kyphosis and lumbar lordosis. This disproportion may indicate that spine disorder also affects other mechanisms related to the control of CoP displacement and suggests that further analyses of a larger number of variables and their interrelationships are warranted in future.

The clear relationship between the reduction in the kyphosis angle and the shortening of the CoG displacement path arises may be due to the fact that postural dysfunction determines the extent of postural reaction disorders, as described by Nault et al. [42]. Rougier et al. [46] explains this phenomenon by the fact that CoG displacement during gait is changed by the forces transmitted to the feet and are partly reduced by compensations occurring in the hip and ankle.

Winter et al. [32] also points out that, with the restoration of the physiological value of thoracic kyphosis, an improvement in balance can be expected, due to shifts in the center of mass in the upper torso, which is then transferred to the ground by the hip and ankle. In contrast, Carlsöö improved dynamic balance by restoration of the physiological size of thoracic kyphosis, which was attributed to the reduction of tensions flowing through the sacro-dorsal and calf muscles to the feet [31].

Our study showed a significant relationship between the angle of kyphosis and the displacement of the CoP in the sagittal plane ($r = 0.438$, $p < 0.001$) and in the coronal plane ($r = 0.182$, $p = 0.008$) (Table 3). With the increase of this curvature, the deviations in the CoP displacement in both planes increased. In their report, Nault et al. presented similar observations, although their research showed a markedly greater effect on the CoP path-length in the coronal plane (increased by 50%) than in the sagittal plane (increased by 25%) [42]. Another report demonstrated that changes in balance parameters correlated with the change in the angle of lumbar lordosis, negating the effect of the angle of thoracic kyphosis [47]. Drzal-Grabiec et al., who conducted research using the formetric system, obtained a significant correlation between both the total CoP path-length and its maximum deviations, depending on the angles of thoracic kyphosis and lumbar lordosis [48], confirming the results obtained in our study.

Maintaining body balance is due to coordinated interaction of the balance organs with the cerebellar and spinal cord centers. Body control in space also requires the interaction of the motor apparatus, eyeballs, and the reticular formation in the brainstem. Adjustment reactions consist of the coordination of tonic-reflex, striving to maintain the body in a vertical position [49]. Atrial-spinal reflexes are the foundation of appropriate postural reactions of the muscles responsible for the position of the neck, torso, and lower limbs. The sensorimotor performance of the musculoskeletal system in this way determines the appropriate equivalent reactions [32].

As the dysfunction increases, postural lengthening is observed in the given CoG, indicating that equivalent reactions are weakened [32]. There is a significant relationship between body posture, efficiency of equivalent reactions, and gait quality [33]. In the case of minor spinal deformations, this may mainly affect the sagittal plane, but with pronounced curvatures of the spine, the dysfunctions also include movement of the CoG in the frontal plane [42].

Body posture is inextricably linked to the ergonomics of gait, and its symmetry is necessary for the proper movement of the CoG during human movement [32]. Movement of the CoG is registered by pressure receptors in the feet, and related reactions aimed at maintaining balance occur by reflex [50].

Our findings confirmed the relationship between body posture disorders and equivalent reactions. Abnormalities in the sagittal plane, i.e., the extent of thoracic kyphosis and lumbar lordosis (Tables 3 and 4) have a particularly marked impact on the increase in the CoP deflection. It is known that disorders in the sagittal plane most commonly underlie dysfunctions in the coronal and transverse planes [51].

The key element of the present study is the method used for assessing spinal curvature and length of CoP displacement. We showed that, when assessing the curvature of the spine in an anatomical manner, the compensatory reactions were clearly dependent on the extent of spinal curvature [47, 48], while no relationship was found when assessing the spinal curvature using the Spinal Maus [52], a device guided manually along the spine on the skin to assess the

curvatures of the vertebral column. The method used for spinal examination appears to be of great importance, as mentioned by Kluszczyński et al. [53]. It is possible that another way of assessing spinal curvature and the displacement of the CoP may lead to other conclusions, which should be evaluated in future research.

## Conclusions

1. In this study, we demonstrated that the extent of kyphosis and lordosis is related to the amplitude of CoP displacement. This association may reflect the postural control system's response to biomechanical destabilisation resulting from altered kyphosis and lordosis.

2. The lordosis angle correlation strength for displacement of COP in sagittal plane is 0.999 and adopts a linear value.

## Supporting information

**S1 Data.**
(ZIP)

## Author Contributions

**Conceptualization:** Arkadiusz Łukaz Żurawski, Wojciech Piotr Kiebzak, Zbigniew Śliwiński.

**Data curation:** Arkadiusz Łukaz Żurawski, Grzegorz Śliwiński.

**Formal analysis:** Arkadiusz Łukaz Żurawski, Ireneusz M. Kowalski, Zbigniew Śliwiński.

**Funding acquisition:** Arkadiusz Łukaz Żurawski, Wojciech Piotr Kiebzak.

**Investigation:** Arkadiusz Łukaz Żurawski, Wojciech Piotr Kiebzak, Zbigniew Śliwiński.

**Methodology:** Arkadiusz Łukaz Żurawski, Wojciech Piotr Kiebzak, Grzegorz Śliwiński, Zbigniew Śliwiński.

**Project administration:** Arkadiusz Łukaz Żurawski, Wojciech Piotr Kiebzak, Ireneusz M. Kowalski, Zbigniew Śliwiński.

**Resources:** Arkadiusz Łukaz Żurawski, Grzegorz Śliwiński.

**Software:** Grzegorz Śliwiński.

**Supervision:** Wojciech Piotr Kiebzak, Ireneusz M. Kowalski, Zbigniew Śliwiński.

**Validation:** Arkadiusz Łukaz Żurawski, Wojciech Piotr Kiebzak, Ireneusz M. Kowalski, Grzegorz Śliwiński, Zbigniew Śliwiński.

**Visualization:** Arkadiusz Łukaz Żurawski, Ireneusz M. Kowalski, Grzegorz Śliwiński, Zbigniew Śliwiński.

**Writing – original draft:** Arkadiusz Łukaz Żurawski, Wojciech Piotr Kiebzak, Zbigniew Śliwiński.

**Writing – review & editing:** Arkadiusz Łukaz Żurawski, Wojciech Piotr Kiebzak, Ireneusz M. Kowalski, Grzegorz Śliwiński, Zbigniew Śliwiński.

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
