## [Decision Letter · Decision Letter 0]

19 May 2020

PONE-D-20-08811

Evaluation of the Efficiency of Compensation Reactions in Children by the Extent of Thoracic Kyphosis and Lumbar Lordosis.

PLOS ONE

Dear Dr Żurawski,

Thank you for submitting your manuscript to PLOS ONE. After careful consideration, we feel that it has merit but does not fully meet PLOS ONE’s publication criteria as it currently stands. Therefore, we invite you to submit a revised version of the manuscript that addresses the points raised during the review process.

We would appreciate receiving your revised manuscript by Jul 03 2020 11:59PM. To enhance the reproducibility of your results, we recommend that if applicable you deposit your laboratory protocols in protocols.io, where a protocol can be assigned its own identifier (DOI) such that it can be cited independently in the future. For instructions see: http://journals.plos.org/plosone/s/submission-guidelines#loc-laboratory-protocols

We look forward to receiving your revised manuscript.

Kind regards,

James G. Wright

Academic Editor

PLOS ONE

Journal Requirements:

2. Please include in your methods section a brief description of how participants were recruited.

Additional Editor Comments (if provided):

Follow recommendations of reviewers closely. Focus manuscript on specific aims. Have manuscript read by native English speaker and use terms that have specific meaning in bio mechanical literature

Reviewers' comments:

Reviewer's Responses to Questions

**Comments to the Author**

1. Is the manuscript technically sound, and do the data support the conclusions?

Reviewer #1: Partly

Reviewer #2: Partly

2. Has the statistical analysis been performed appropriately and rigorously? 

Reviewer #1: Yes

Reviewer #2: No

3. Have the authors made all data underlying the findings in their manuscript fully available?

Reviewer #1: Yes

Reviewer #2: No

4. Is the manuscript presented in an intelligible fashion and written in standard English?

Reviewer #1: No

Reviewer #2: No

5. Review Comments to the Author

Reviewer #1: The authors present a manuscript about compensation reactions in children related to the amount of thoracic kyphosis and lumbar lordosis. While their manuscript contained interesting information, it could not be approved as submitted. If the authors choose to revise their manuscript, please address the following comments. As a general advice, please have a native English speaker revise your manuscript in order to enhance its readability:

Page 2, Abstract: “The study group consists of 211 patients with postural disorders with respect to the degree of thoracic kyphosis and lumbar lordosis…” Please define what you mean by postural disorders: hyperkyphosis, hypokyphosis, hyperlordosis etc.?

Page 2, Abstract: “Along with the increase in the angle of lordosis, an increase in indicators was observed: maximum movement of the CoP in the coronal plane, maximum movement of the CoP to the side (statically body to the left); and a reduction in the forward movement of the maximum CoP in the static test and the backward movement of the maximum CoP in the static test.” Please clarify.

Introduction I suggest that you group together the results of the different studies you mentioned. The Wenger and Frick study should be put apart as it points to the fundamental fact that sagittal spinal alignment is age dependent, which is important for your study

Page 4 “The main element in controlling body posture is balance. Its typical parameters are constituted of measures of the body’s vertical orientation, obtained by balancing out the forces acting on different body segments [17]. This phenomenon is related to inertia forces and inertia characteristics of individual body segments [18].” This information is considered common knowledge. For the sake of concision, I suggest omitting these phrases.

Page 4 “The determinant used to define the ability to maintain balance are the center of pressure (CoP) movements, the establishing of which allows controlling the treatment process…” What do you mean by “allows controlling the treatment process”? Please clarify

Page 4 “With the increase in postural dysfunction, an extension of the path covered by the centre of pressure in a given unit of time is observed, which means that compensating reactions are weakened [18].” Please be more explicit. As this is mean subject of your research project, this deserves more explanations

Page 4 “postural muscle disbalance” consider muscle imbalance

Page 4 “unnormal” = abnormal

Purpose of the study “This study aims to evaluate the efficiency of compensating reactions in children by the extent of thoracic kyphosis and lumbar lordosis…” my proposal: Our project is to study the influence of thoracic kyphosis and lumbar lordosis on compensating reactions in children?

Page 5 “body posture defect observed in clinical trial related to abnormalities in thoracic kyphosis and lumbar lordosis” Please give us the threshold above or below which the thoracic kyphosis and lumbar lordosis were considered abnormal

Page 6 Please provide the references of the “DIERS formetric 4D system”

Discussion

Page 10 “The inspirational aspect lies in the fact that body posture disorders make up one of the factors that can affect various systems of the human body [31], starting with pain accompanying postural dysfunctions [32-35], through pulmonary ventilation disorders [36], intestinal motility disorders to malocclusion [35]…” Those comments are controversial and stray away from the main subject of this work. It will induce some readers to question the seriousness of this work and might induce them to stop reading this article before apprehending the full extent of your work. My advice: omit this phrase…

Page 11 “while no relationship was found when assessing the figure using Spinal Maus [40]…” The average reader might not be aware about the signification of the “Spinal Maus” ? Please explain

Clinical Significance This chapter contains some general comments which don’t provide significant information. I would suggest repeating with simple words the result of your research: “Increased sagittal curves of the spine provoke increased movements of the center of gravity.” How this relates to clinical problems should be the subject of another study.

Conclusions You mention conclusion number 3 and 4: what are the other conclusions (number 1 and 2)?

Reviewer #2: The following revisions are recommended:

General: overall the manuscript presents interesting findings. However, it is recommended that the authors review the manuscript, and incorporate significant revisions. The major revisions that are required are:

1) Analysis: include more detail in the results, and also statistical analysis between groups.

2) Re-write the objectives to more directly reflect the analysis.

3) Re-vise/re-organize/re-write some sections in the introduction/discussion: This is important, as the strength of the data set is lost in it's current form, as the message is difficult to follow. (see below)

1. Abstract:

- please clarify 'The determinant used to define the ability to maintain balance are the center of pressure (CoP) movements, the establishing of which allows controlling the treatment process.' What do you mean by controlling the treatment process. Recommend separating this sentence into two parts, so that the 'treatment' implication is more clearly described.

Objective: please see above.

Methods:please consider being more specific in relation to the term postural disorder and spinal curvature. The context of the study is postural disorders, but the measurements obtained are specific to spinal curvature which are one component of a postural disorder. (see comments below for introduction)

- results: rather than using the term 'equilibrium reactions', suggest using the term COP displacement.

Conclusion:

conclusion 1: suggest reporting on what was observed statistically, I don't agree that the size of the curve alone resulted in large equilibrium reaction. Rather there is a positive association between kyphosis, lordosis and amplitude of COP displacement. This association is a potential reflection of the postural control's system response to the structural/mechanical destabilisation that is caused by increased kyphosis/lordosis.

conclusion 2: is this not included in conclusion 1? suggest revising this 2nd conclusion, and being more specific in relation to the strength of the association, and not just the linearity.

2. Introduction:

1st paragraph: this is an important paragraph to provide the overall context of the present study. It is suggested that the author's include a more specific description of what is involved with a postural disorder. For example, the sentence 'A faulty posture comprises a set of irregularities, defined as small, individual deviations from the correct posture, which can be corrected using a properly selected therapy'.... can be further described by including reference to the underlying skeletal structures that impact a correct alignment: (ie., lower limb alignment, pelvis position/orientation, spine/thoracic cage, scapular position etc.....' from this, then indicate that there is an interest in further understanding how a spinal alignment (thorcic, lumbar) is assessed and treated.

2nd paragraph: In consideration above, then be more specific in the first sentence, and introduce kyphosis and lordosis within the context of spinal alignment.

3rd paragraph: suggest revising this paragraph.

Sentence 1: 'The main element in controlling body posture is balance.' This is to general of a statement. It is recommended to be more specific in this paragraph to draw upon the link between postural alignment and the CNS driven mechanisms that the body utilises to control upright posture. The next two sentences are clearer, and more specific, thus it is suggested to start this paragraph with these two sentences, and introduce the idea that the CNS attempts to counteract the gravitational forces that are acting on the body by adjusting alignment of body segments (compensation), and secondly generating postural responses to control this alignment. A measure that reflects these postural responses is the centre of pressure, which is a reflection of the net torque generated about each body segment.

3. Purpose: see above comment, it is suggested to be more specific and state 2 objectives that better link with the statistical anlaysis. (ie., 1: compare between groups; 2: explore associations.

4. Methods:

- general comment: revise the decimal precision to be consistent (ie., to 1 decimal point).

- Inclusion/exclusion criteria: it is not clear what the threshold was for defining a postural defect, please define what is ECOG/WHO.

- how was movement of the COP quantified, and what was the actual measure? Was it standard deviation, root mean square, peak max, min, range. This has to be more clearly specified.

5. Results:

General: multiple correlations are performed. Did the authors adjust the p value for multiple comparisons?

Paragraph 5: what is the statistical comparison between the groups.

Discussion: It is suggested to re-organize the discussion with a specific focus on the following:

Current Paragraphs:

Paragraph 1: this interpretation should be moved later in the discussion, and move content/ideas to suggested paragraph 5 (see below)

Paragraph 2: this interpretation should be moved later in the discussion, and move content/ideas to suggested paragraph 5(see below)

Paragraph 3: integrate this into the suggested new paragraph 4 (see below).

Paragraph 4: include this in recommended paragraph 2 (see below)

Paragraph 5: include this in recommended paragraph 6 (see below). Mainly, this is a methodological discussion, which relates to the different techniques that are used to predict spinal curvature from the surface of the skin. Thus, the validity of the current technique should be discussed in relation to other techniques.

Paragraph 6: remove, or integrate into paragraph 5 below.

Paragraph 7: remove, or integrate into paragraph 5 below.

Paragraph 8: clinical significance: integrate into paragraph 5 below

New organization

Paragraph 1: Restate the primary objectives, and the main findings.

Paragraph 2 and 3: Compare the findings obtained in the present study with those of previous literature. This comparison is made in the current 3rd paragraph. It is recommended to move this up to 2nd paragraph, and expand upon this comparison. First, compare the findings to previous work with respect to 1:static posture; 2) walking. Also include specific comments related to 1) differences between the clinical population and the non-clinical population; 2) the strength of the association between kyphosis/lordosis and the COP displacement (in static, and dynamic).

Paragraph 4: Interpret your findings in relation to the population differences, and also the relationships between measures. This should specifically focus on the relationship between COP excursion and kyphosis/lordosis. Bring into consideration what the COP reflects from a CNS/musculoskeletal/torque perspective, and how structural alignment of the spine affects the mechanical stability of the body. For example, this reviewer suggests the following lines of thought/interpretation:

CNS compensates for structural alignment of the body segments. IN this case, there seems to be a relationship between COP and amplitude of kyphosis or lordosis. Is this increase in COP excursion a reflection of 1) mechanical detabilisation of the trunk caused by greater kyphosis/lordosis; 2) response of CNS to modulate the torques at the ankle,hips (during standing), and ankle, knee, hips during walking. This greater modulation of torque is then reflected through increased COP excursion in static/dynamic tasks.

Paragraph 5: interpretation within the broader context: what do these findings imply with respect to clinical assessment and management of patients.

Paragraph 6: limitations:

- consider methodological aspects that you brought into the discussion in paragraph 5(see above). this relates to the validity/reliability of the surface model to predict kyphosis/lordosis.

- consider the internal and external threats to validity.

Conclusion: revise conclusion.

- there is not a cause-effect shown in this study, rather there is an association that is demonstrated.

- see comments above for conclusion in the abstract.

6. PLOS authors have the option to publish the peer review history of their article (what does this mean?). If published, this will include your full peer review and any attached files.

Reviewer #1: No

Reviewer #2: No

---

## [Author Response · Author response to Decision Letter 0]

8 Jul 2020

Dear Editor and dear reviewers,

Thank you for your thorough assessment and valuable tips on the complex work, entitled: "Efficiency of Compensation Reactions in Children by the Extent of Thoracic Kyphosis and Lumbar Lordosis ". 

We are of the opinion that all suggestions given are correct and their application will positively influence the comprehensibility of the text and clarity of the message. Therefore, during changes in the text, we tried to apply as closely as possible all the comments provided.

In the remainder of the letter I will try to answer in detail how the individual suggestions were responded, in the order in which they were submitted.

Answers to editor's comments:

Answer: The professional editing company took care of adjusting the text corrected by the reviewers, so the style of the corrected text sent is in line with PLOS ONE requirements.

2. Please include in your methods section a brief description of how participants were recruited.

Answer: The paragraph describing the recruitment process of participants was expanded. 

3. Have manuscript read by native English speaker and use terms that have specific meaning in bio mechanical literature.

Answer: The language edition was commissioned to a professional company, which takes into account the specificity of the thematic area and the language requirements of PLOS ONE during the correction.

Answers to Reviewer #1

The authors present a manuscript about compensation reactions in children related to the amount of thoracic kyphosis and lumbar lordosis. While their manuscript contained interesting information, it could not be approved as submitted. If the authors choose to revise their manuscript, please address the following comments. As a general advice, please have a native English speaker revise your manuscript in order to enhance its readability.

Answer: Thank you to the reviewer for your time and valuable comments. We appreciate that he noticed the value of the sent manuscript. We apologize for any language errors. We commissioned the language correction of the corrected text to a professional company and we hope that the language of the corrected version does not raise any objections.

Page 2, Abstract: “The study group consists of 211 patients with postural disorders with respect to the degree of thoracic kyphosis and lumbar lordosis…” Please define what you mean by postural disorders: hyperkyphosis, hypokyphosis, hyperlordosis etc.?

Answer: The reviewer rightly noted that the term used was not precise. We have improved it both in the summary and in the part describing the material and the research method. 

Page 2, Abstract: “Along with the increase in the angle of lordosis, an increase in indicators was observed: maximum movement of the CoP in the coronal plane, maximum movement of the CoP to the side (statically body to the left); and a reduction in the forward movement of the maximum CoP in the static test and the backward movement of the maximum CoP in the static test.” Please clarify.

Introduction I suggest that you group together the results of the different studies you mentioned. The Wenger and Frick study should be put apart as it points to the fundamental fact that sagittal spinal alignment is age dependent, which is important for your study

Answer: The results described in the summary have been described more clearly

Page 4 “The main element in controlling body posture is balance. Its typical parameters are constituted of measures of the body’s vertical orientation, obtained by balancing out the forces acting on different body segments [17]. This phenomenon is related to inertia forces and inertia characteristics of individual body segments [18].” This information is considered common knowledge. For the sake of concision, I suggest omitting these phrases.

Answer: As suggested, this part has been removed.

Page 4 “The determinant used to define the ability to maintain balance are the center of pressure (CoP) movements, the establishing of which allows controlling the treatment process…” What do you mean by “allows controlling the treatment process”? Please clarify

Answer: We apologize for too vague returns. We have expanded this sentence. 

Page 4 “With the increase in postural dysfunction, an extension of the path covered by the centre of pressure in a given unit of time is observed, which means that compensating reactions are weakened [18].” Please be more explicit. As this is mean subject of your research project, this deserves more explanations

Answer: Thank you for your valuable attention. We changed this part. Page 4 “postural muscle disbalance” consider muscle imbalance

Page 4 “unnormal” = abnormal

Answer: Thank you for the suggestion. As a suggestion, both phrases have been replaced with those proposed throughout the text.

Purpose of the study “This study aims to evaluate the efficiency of compensating reactions in children by the extent of thoracic kyphosis and lumbar lordosis…” my proposal: Our project is to study the influence of thoracic kyphosis and lumbar lordosis on compensating reactions in children?

Answer: Thank you for the suggestion. Reviewer 2 also commented on this part, so taking both suggestions into account the sentence " This study evaluated the efficiency of compensatory reactions in children according to the extent of thoracic kyphosis and lumbar lordosis.".

Page 5 “body posture defect observed in clinical trial related to abnormalities in thoracic kyphosis and lumbar lordosis” Please give us the threshold above or below which the thoracic kyphosis and lumbar lordosis were considered abnormal.

Answer: The reference ranges for the size of thoracic kyphosis and lumbar lordosis were determined on the basis of the work suggested by the device manufacturer describing large screening tests. This information was included in the text. 

Page 6 Please provide the references of the “DIERS formetric 4D system”

Answer: The paper was supplemented with suggested references for research on the DIERS system. Publications cited in the manuscript are numbered: 7,35,36,37,38,39,40 in the literature.

Page 10 “The inspirational aspect lies in the fact that body posture disorders make up one of the factors that can affect various systems of the human body [31], starting with pain accompanying postural dysfunctions [32-35], through pulmonary ventilation disorders [36], intestinal motility disorders to malocclusion [35]…” Those comments are controversial and stray away from the main subject of this work. It will induce some readers to question the seriousness of this work and might induce them to stop reading this article before apprehending the full extent of your work. My advice: omit this phrase…

Answer: As suggested, this part has been removed.

Page 11 “while no relationship was found when assessing the figure using Spinal Maus [40]…” The average reader might not be aware about the signification of the “Spinal Maus” ? Please explain.

Answer: This sentence indicates a difference in measurements that take place automatically and when manual participation of the researcher is needed. The text added the sentence: " This device is guided manually on the skin along the spine to assesses the curvatures of the vertebral column".

Clinical Significance This chapter contains some general comments which don’t provide significant information. I would suggest repeating with simple words the result of your research: “Increased sagittal curves of the spine provoke increased movements of the center of gravity.” How this relates to clinical problems should be the subject of another study.

Answer: This part of the text has been changed as suggested and linked to the discussion. 

Conclusions You mention conclusion number 3 and 4: what are the other conclusions (number 1 and 2)?

Answer: The conclusions in the manuscript sent were incorrectly numbered. Currently, after the suggestions of Reviewer 2, this part of the manuscript looks as follows:

 "1. We found an association between kyphosis and lordosis and the amplitude of CoP displacement, which may reflect the postural control system’s response to biomechanical destabilisation caused by changes in kyphosis and lordosis.

2. The lordosis angle correlation strength for displacement of COP in sagittal plane is 0.999 and adopts a linear value.".

Answers for Reviewer 2.

 Thank you for the extremely detailed analysis of the manuscript submitted. We are grateful for the enormity of our time manuscript. In the process of improving the manuscript, we followed all comments, which resulted in a complete rebuilding of the manuscript. I will refer in turn to all suggested and introduced changes.

General: overall the manuscript presents interesting findings. However, it is recommended that the authors review the manuscript, and incorporate significant revisions. The major revisions that are required are:

1) Analysis: include more detail in the results, and also statistical analysis between groups.

2) Re-write the objectives to more directly reflect the analysis.

3) Re-vise/re-organize/re-write some sections in the introduction/discussion: This is important, as the strength of the data set is lost in it's current form, as the message is difficult to follow. (see below)

Answer: 

Ad.1 A table was added comparing the results of the parameters obtained for both groups.

Ad2. According to the reviewer's recommendations, the following was used: „This study evaluated the efficiency of compensatory reactions in children according to the extent of thoracic kyphosis and lumbar lordosis.”.

Ad.3 In principle, the entire manuscript is reorganized to stick to the suggestions.

1. Abstract:

- please clarify 'The determinant used to define the ability to maintain balance are the center of pressure (CoP) movements, the establishing of which allows controlling the treatment process.' What do you mean by controlling the treatment process. Recommend separating this sentence into two parts, so that the 'treatment' implication is more clearly described.

Answer: The following explanatory note has been added.

Objective: please see above.

Answer: Changed as suggested by the reviewer to: " This study evaluated the efficiency of compensatory reactions in children according to the extent of thoracic kyphosis and lumbar lordosis.".

Methods:please consider being more specific in relation to the term postural disorder and spinal curvature. The context of the study is postural disorders, but the measurements obtained are specific to spinal curvature which are one component of a postural disorder. (see comments below for introduction)

Answer: The reference ranges for the size of thoracic kyphosis and lumbar lordosis were determined on the basis of the work suggested by the device manufacturer describing large screening tests. This information was included in the text

Results: rather than using the term 'equilibrium reactions', suggest using the term COP displacement.

Answer: Changed as directed

Conclusion:

conclusion 1: suggest reporting on what was observed statistically, I don't agree that the size of the curve alone resulted in large equilibrium reaction. Rather there is a positive association between kyphosis, lordosis and amplitude of COP displacement. This association is a potential reflection of the postural control's system response to the structural/mechanical destabilisation that is caused by increased kyphosis/lordosis.

conclusion 2: is this not included in conclusion 1? suggest revising this 2nd conclusion, and being more specific in relation to the strength of the association, and not just the linearity.

Answer: As suggested, the applications were reworded. 

2.Introduction:

1st paragraph: this is an important paragraph to provide the overall context of the present study. It is suggested that the author's include a more specific description of what is involved with a postural disorder. For example, the sentence 'A faulty posture comprises a set of irregularities, defined as small, individual deviations from the correct posture, which can be corrected using a properly selected therapy'.... can be further described by including reference to the underlying skeletal structures that impact a correct alignment: (ie., lower limb alignment, pelvis position/orientation, spine/thoracic cage, scapular position etc.....' from this, then indicate that there is an interest in further understanding how a spinal alignment (thorcic, lumbar) is assessed and treated.

Answer: As suggested, the following information has been added.

2nd paragraph: In consideration above, then be more specific in the first sentence, and introduce kyphosis and lordosis within the context of spinal alignment.

Answer: As suggested, the following information has been added.

3rd paragraph: suggest revising this paragraph.

Sentence 1: 'The main element in controlling body posture is balance.' This is to general of a statement. It is recommended to be more specific in this paragraph to draw upon the link between postural alignment and the CNS driven mechanisms that the body utilises to control upright posture. The next two sentences are clearer, and more specific, thus it is suggested to start this paragraph with these two sentences, and introduce the idea that the CNS attempts to counteract the gravitational forces that are acting on the body by adjusting alignment of body segments (compensation), and secondly generating postural responses to control this alignment. A measure that reflects these postural responses is the centre of pressure, which is a reflection of the net torque generated about each body segment.

Answer: As suggested, this paragraph was rebuilt.

3. Purpose: see above comment, it is suggested to be more specific and state 2 objectives that better link with the statistical anlaysis. (ie., 1: compare between groups; 2: explore associations.

Answer: Taking into account your suggestion as well as the suggestion of the first reviewer, the purpose is described as follows: „This study evaluated the efficiency of compensatory reactions in children according to the extent of thoracic kyphosis and lumbar lordosis.”.

4. Methods:

- general comment: revise the decimal precision to be consistent (ie., to 1 decimal point).

- Inclusion/exclusion criteria: it is not clear what the threshold was for defining a postural defect, please define what is ECOG/WHO.

- how was movement of the COP quantified, and what was the actual measure? Was it standard deviation, root mean square, peak max, min, range. This has to be more clearly specified.

Answer: As suggested, this part has been corrected.

5. Results:

General: multiple correlations are performed. Did the authors adjust the p value for multiple comparisons?

Paragraph 5: what is the statistical comparison between the groups.

Answer: Tables were added comparing the size of parameters in the control and study groups. This is the current version of the manuscript table number 3. During statistical analysis, no protocol was used to adjust the p-value for many comparisons (such as the Dunn test). This is due to the fact that 6 correlations were generally tested in the study group. The same correlations were tested in the control group, and the parameters in the control group were compared. We realize that this can be understood as testing 12 correlations and then the use of p-value adjustment may be justified. However, low-strength correlations have only been signaled, and the work mainly focuses on very strong correlations that take linear values.

Discussion: It is suggested to re-organize the discussion with a specific focus on the following:

Current Paragraphs:

Paragraph 1: this interpretation should be moved later in the discussion, and move content/ideas to suggested paragraph 5 (see below)

Paragraph 2: this interpretation should be moved later in the discussion, and move content/ideas to suggested paragraph 5(see below)

Paragraph 3: integrate this into the suggested new paragraph 4 (see below).

Paragraph 4: include this in recommended paragraph 2 (see below)

Paragraph 5: include this in recommended paragraph 6 (see below). Mainly, this is a methodological discussion, which relates to the different techniques that are used to predict spinal curvature from the surface of the skin. Thus, the validity of the current technique should be discussed in relation to other techniques.

Paragraph 6: remove, or integrate into paragraph 5 below.

Paragraph 7: remove, or integrate into paragraph 5 below.

Paragraph 8: clinical significance: integrate into paragraph 5 below

New organization

Paragraph 1: Restate the primary objectives, and the main findings.

Paragraph 2 and 3: Compare the findings obtained in the present study with those of previous literature. This comparison is made in the current 3rd paragraph. It is recommended to move this up to 2nd paragraph, and expand upon this comparison. First, compare the findings to previous work with respect to 1:static posture; 2) walking. Also include specific comments related to 1) differences between the clinical population and the non-clinical population; 2) the strength of the association between kyphosis/lordosis and the COP displacement (in static, and dynamic).

Paragraph 4: Interpret your findings in relation to the population differences, and also the relationships between measures. This should specifically focus on the relationship between COP excursion and kyphosis/lordosis. Bring into consideration what the COP reflects from a CNS/musculoskeletal/torque perspective, and how structural alignment of the spine affects the mechanical stability of the body. For example, this reviewer suggests the following lines of thought/interpretation:

CNS compensates for structural alignment of the body segments. IN this case, there seems to be a relationship between COP and amplitude of kyphosis or lordosis. Is this increase in COP excursion a reflection of 1) mechanical detabilisation of the trunk caused by greater kyphosis/lordosis; 2) response of CNS to modulate the torques at the ankle,hips (during standing), and ankle, knee, hips during walking. This greater modulation of torque is then reflected through increased COP excursion in static/dynamic tasks.

Paragraph 5: interpretation within the broader context: what do these findings imply with respect to clinical assessment and management of patients.

Paragraph 6: limitations:

- consider methodological aspects that you brought into the discussion in paragraph 5(see above). this relates to the validity/reliability of the surface model to predict kyphosis/lordosis.

- consider the internal and external threats to validity.

Answer: The entire discussion was reorganized according to the detailed guidelines.

Conclusion: revise conclusion.

- there is not a cause-effect shown in this study, rather there is an association that is demonstrated.

- see comments above for conclusion in the abstract.

Answer: As suggested, the conclusions have been improved in both the summary and the main part. The conclusions now read as follows:

„1. We found an association between kyphosis and lordosis and the amplitude of CoP displacement, which may reflect the postural control system’s response to biomechanical destabilisation caused by changes in kyphosis and lordosis.

2. The lordosis angle correlation strength for displacement of COP in sagittal plane is 0.999 and adopts a linear value”.

 Thank you for the opportunity to make changes in our manuscript. We have referred to all your comments and suggestions. We believe, that in current form, our manuscript meets the standards of PLOS ONE.

Sincerely,

the Authors.

---

## [Decision Letter · Decision Letter 1]

29 Sep 2020

PONE-D-20-08811R1

Evaluation of the Efficiency of Compensation Reactions in Children by the Extent of Thoracic Kyphosis and Lumbar Lordosis.

PLOS ONE

Dear Dr. Żurawski,

Thank you for submitting your manuscript to PLOS ONE. After careful consideration, we feel that it has merit but does not fully meet PLOS ONE’s publication criteria as it currently stands. Therefore, we invite you to submit a revised version of the manuscript that addresses the points raised during the review process.

We look forward to receiving your revised manuscript.

Kind regards,

James G. Wright

Academic Editor

PLOS ONE

Additional Editor Comments (if provided):

Considerable improvement in response to first review, further explanations are requested

Reviewers' comments:

Reviewer's Responses to Questions

**Comments to the Author**

1. If the authors have adequately addressed your comments raised in a previous round of review and you feel that this manuscript is now acceptable for publication, you may indicate that here to bypass the “Comments to the Author” section, enter your conflict of interest statement in the “Confidential to Editor” section, and submit your "Accept" recommendation.

Reviewer #1: All comments have been addressed

Reviewer #2: (No Response)

2. Is the manuscript technically sound, and do the data support the conclusions?

Reviewer #1: Yes

Reviewer #2: Partly

3. Has the statistical analysis been performed appropriately and rigorously? 

Reviewer #1: Yes

Reviewer #2: Yes

4. Have the authors made all data underlying the findings in their manuscript fully available?

Reviewer #1: Yes

Reviewer #2: Yes

5. Is the manuscript presented in an intelligible fashion and written in standard English?

Reviewer #1: Yes

Reviewer #2: No

6. Review Comments to the Author

Reviewer #1: The authors undertook a comprehensive review of their article and addressed my comments in a satisfactory way. Their manuscript is now better structured and easier to read. It contains valuable information and therefore I recommend this manuscript for publication.

However, there still remains a minor critical comment. I would still suggest omitting the following phrases as they stray away from the main theme and contain controversial statements:

Page 16: Body posture disorders can affect various systems of the human body [52], from the

pain accompanying postural dysfunctions [53-56], through pulmonary ventilation disorders

[57] and intestinal motility disorders, to malocclusion [54]. The relationship between spinal

curvature and balance disorders has been confirmed in numerous studies [29, 30, 54, 58, 59].

Page 17: Most authors use the Romberg test to assess body balance. However, the ability to use vision to improve motor skills in children during development is only emerging and both values can be aligned closely [61].

Reviewer #2: Thank you for revising the manuscript and incorporating the suggested revisions. The following revisions are further recommended:

1. Title: Revise the title, as it is not clear. It is suggested that the revised title more closely reflect the analysis. For example'Evaluation of the association between postural control and sagittal curvature of the spine'.

2. Abstract: revise the text:

‘worse’ with ‘smaller’

‘coronary’ with coronal

3. Introduction:

1. Revise: ‘ The physiological values…’ to ‘ The amplitude of ‘

2. Combine 2nd and 3rd paragraphs. ‘ Body posture disorders…..’ with the next paragraph starting with ‘The literature’….

3. It is important to better distinguish the displacement of the COP under both feet during upright quiet standing, with that of the COP displacement under 1 foot during walking. Although this is described in paragraph 4, it is recommended to revise this paragraph to be more clear.

For example:

1. COP displacement during quiet standing reflects the net contribution of the CNS to control the movement of the Centre of Mass. Where the horizontal acceleration of the COM is proportional to the difference between the COP-COM. (Winter et al., ) during quiet stance it is the hip abductors/adductors that generate torque to control ML movement, and it is the ankle plantar flexors/dorsiflexors that control A/P movement

2. During walking, the movement of the COP moves from posterior to anterior under the foot, and ML. Although the above concept holds true (ie., COP reflects net output of CNS control), how it does this is more complex given the multiple degrees of freedom, and joint motions.

In light of this, it is suggested to revise this paragraph to further strengthen the rational as to why the investigators chose the COP as the primary outcome variable to measure standing stability, and also stability during walking.

Methods:

1. Clarify for how long the centre of pressure was recorded. Was it 3 seconds, the same as the rastersterography, or a different length of time.

2. Research methodology: It is recommended to define each parameter that was included in the analysis. For example, Table 2 contains variables such as ‘Maximum displacement of centre of gravity in coronal plane in pressure gait’. This is very difficult to understand what this variable is in the table. Therefore, each measure described in Table 2 should be defined in the methods.

a. Angle of Thoracic Kyphsosi, Angle of lumbar Lordosis

b. Displacement of centre of gravity in coronal/sagittal planes in pressure gait

c. Maximum displacement of centre of pressure to side [mm].

Results:

1. Table 2, 3 and 4 are redundant. It is recommended to include primarily Table 4.

2. Revise the variable names to be consistent with the definition in the methods, and also how COP displacement is referred to throughout the manuscript. For example:

a. ‘Maximum displacement of center of gravity in coronal plane in pressure gait’. Suggest revising to ‘Maximum displacement of centre of pressure in coronal plane during gait’.

b. ‘Maximum displacement of center of gravity in sagittal plane in pressure gait’. Suggest revising to ‘Maximum displacement of centre of pressure in sagittal plane during gait’.

Discussion

- It is suggested to use consistent terminology with respect to COP displacement. Rather than using terms such as ‘swing’, ‘deflections’, it is suggested to consistently the word ‘displacement’ throughout the text.

- Revise – Nalut, and Naulut to correct reference ‘ Nault’.

- Revise statement ‘ Rougier et al., explains this phenomenon by the fact that swing during walking is changed by the forces tramsitted to the feet and are partly reduced by compensations occurring in the hip and ankle’. It is not clear what ‘swing’ means. Is this the swing phase of gait?

7. PLOS authors have the option to publish the peer review history of their article (what does this mean?). If published, this will include your full peer review and any attached files.

Reviewer #1: **Yes: **Reinhard Zeller

Reviewer #2: No

---

## [Author Response · Author response to Decision Letter 1]

3 Oct 2020

Dear Editor and dear Reviewers,

Thank you for your thorough assessment and valuable tips on the complex work, entitled: "Efficiency of Compensation Reactions in Children by the Extent of Thoracic Kyphosis and Lumbar Lordosis ". 

We are of the opinion that all suggestions given are correct and their application will positively influence the comprehensibility of the text and clarity of the message. Therefore, during changes in the text, we tried to apply as closely as possible all the comments provided.

In the remainder of the letter I will try to answer in detail how the individual suggestions were responded, in the order in which they were submitted.

Answers to Reviewer #1

The authors undertook a comprehensive review of their article and addressed my comments in a satisfactory way. Their manuscript is now better structured and easier to read. It contains valuable information and therefore I recommend this manuscript for publication.

However, there still remains a minor critical comment. I would still suggest omitting the following phrases as they stray away from the main theme and contain controversial statements:

Page 16: Body posture disorders can affect various systems of the human body [52], from the

pain accompanying postural dysfunctions [53-56], through pulmonary ventilation disorders

[57] and intestinal motility disorders, to malocclusion [54]. The relationship between spinal

curvature and balance disorders has been confirmed in numerous studies [29, 30, 54, 58, 59].

Page 17: Most authors use the Romberg test to assess body balance. However, the ability to use vision to improve motor skills in children during development is only emerging and both values can be aligned closely [61].

Answer: Thank you for this suggestion. As suggested, this part has been removed.

Answers for Reviewer 2.

Thank you for revising the manuscript and incorporating the suggested revisions. The following revisions are further recommended:

1. Title: Revise the title, as it is not clear. It is suggested that the revised title more closely reflect the analysis. For example' Evaluation of the association between postural control and sagittal curvature of the spine '.

Answer: Thanks for the suggestion. The title has been changed as proposed.

2. Abstract: revise the text:

‘worse’ with ‘smaller’

‘coronary’ with coronal

Answer: Thanks for the suggestion. Words changed: 'worse' with 'greater', 'coronary' with coronal '. Worse in this case meant a longer diplacement in all directions, so instead of 'worse' we put 'greater'.

3. Introduction:

1. Revise: ‘ The physiological values…’ to ‘ The amplitude of ‘

Thanks for the suggestion. The text has been changed as proposed.

2. Combine 2nd and 3rd paragraphs. ‘ Body posture disorders…..’ with the next paragraph starting with ‘The literature’….

Answer: Thanks for the suggestion. The text has been changed as proposed.

3. It is important to better distinguish the displacement of the COP under both feet during upright quiet standing, with that of the COP displacement under 1 foot during walking. Although this is described in paragraph 4, it is recommended to revise this paragraph to be more clear.

For example:

1. COP displacement during quiet standing reflects the net contribution of the CNS to control the movement of the Centre of Mass. Where the horizontal acceleration of the COM is proportional to the difference between the COP-COM. (Winter et al., ) during quiet stance it is the hip abductors/adductors that generate torque to control ML movement, and it is the ankle plantar flexors/dorsiflexors that control A/P movement

2. During walking, the movement of the COP moves from posterior to anterior under the foot, and ML. Although the above concept holds true (ie., COP reflects net output of CNS control), how it does this is more complex given the multiple degrees of freedom, and joint motions.

In light of this, it is suggested to revise this paragraph to further strengthen the rational as to why the investigators chose the COP as the primary outcome variable to measure standing stability, and also stability during walking.

Answer: Thank you for this suggestion. Section 4 has been supplemented with the proposed content.

Methods:

1. Clarify for how long the centre of pressure was recorded. Was it 3 seconds, the same as the rastersterography, or a different length of time.

Answer: Thank you for this suggestion. the text was supplemented with the sentence 'The measurement was carried out for 3 seconds at the same time as the shape of spine was assessed with the DIERS formetric system.'

2. Research methodology: It is recommended to define each parameter that was included in the analysis. For example, Table 2 contains variables such as ‘Maximum displacement of centre of gravity in coronal plane in pressure gait’. This is very difficult to understand what this variable is in the table. Therefore, each measure described in Table 2 should be defined in the methods.

a. Angle of Thoracic Kyphsosi, Angle of lumbar Lordosis

b. Displacement of centre of gravity in coronal/sagittal planes in pressure gait

c. Maximum displacement of centre of pressure to side [mm].

Answer: Thank you for this valuable suggestion. The parameters were described and their nomenclature arranged throughout the manuscript.

Results:

1. Table 2, 3 and 4 are redundant. It is recommended to include primarily Table 4.

Answer: Thank you for the suggestion. Tables removed as recommended.

2. Revise the variable names to be consistent with the definition in the methods, and also how COP displacement is referred to throughout the manuscript. For example:

a. ‘Maximum displacement of center of gravity in coronal plane in pressure gait’. Suggest revising to ‘Maximum displacement of centre of pressure in coronal plane during gait’.

b. ‘Maximum displacement of center of gravity in sagittal plane in pressure gait’. Suggest revising to ‘Maximum displacement of centre of pressure in sagittal plane during gait’.

Answer: Thank you for these suggestions. The text was corrected as recommended.

Discussion

- It is suggested to use consistent terminology with respect to COP displacement. Rather than using terms such as ‘swing’, ‘deflections’, it is suggested to consistently the word ‘displacement’ throughout the text.

Thank you for these suggestions. The text was corrected as recommended.

- Revise – Nalut, and Naulut to correct reference ‘ Nault’.

Answer: Thank you for these suggestions. The text was corrected as recommended.

- Revise statement ‘ Rougier et al., explains this phenomenon by the fact that swing during walking is changed by the forces tramsitted to the feet and are partly reduced by compensations occurring in the hip and ankle’. It is not clear what ‘swing’ means. Is this the swing phase of gait?

Answer: Thank you for these suggestions. The text was corrected as recommended. The statement 'Rougier et al., Explains this phenomenon by the fact that swing during walking is changed by the forces tramsitted to the feet and are partly reduced by compensations occurring in the hip and ankle ’changed to' Rougier et al. this phenomenon by the fact that CoG displacement during walking is changed by the forces transmitted to explain the feet and are partly reduced by compensations occurring in the hip and ankle '.

 Thank you for the opportunity to make changes in our manuscript. We have referred to all your comments and suggestions. We believe, that in current form, our manuscript meets the standards of PLOS ONE.

Sincerely,

the Authors.

---

## [Editor Report · Decision Letter 2]

12 Oct 2020

Evaluation of the association between postural control and sagittal curvature of the spine

PONE-D-20-08811R2

Dear Dr. Żurawski,

We’re pleased to inform you that your manuscript has been judged scientifically suitable for publication and will be formally accepted for publication once it meets all outstanding technical requirements.

Kind regards,

James G. Wright

Academic Editor

PLOS ONE
---

## [Editor Report · Acceptance letter]

14 Oct 2020

PONE-D-20-08811R2 

Evaluation of the association between postural control and sagittal curvature of the spine 

Dear Dr. Żurawski:

I'm pleased to inform you that your manuscript has been deemed suitable for publication in PLOS ONE. Congratulations! Your manuscript is now with our production department. 

Kind regards, 

on behalf of

Professor James G. Wright 

Academic Editor

PLOS ONE